# Building Agroforestry Policy Bottom-Up: Knowledge of Czech Farmers on Trees in Farmland

**Jana Krčmářová [1],\*, Lukáš Kala [2], Alica Brendzová [1] and Tomáš Chabada [3]**

1   Institute of Ethnology, Czech Academy of Sciences, Na Florenci 3, 110 00 Prague, Czech Republic;
    alica.brendzova@gmail.com
2   The Institute of Botany of the Czech Academy of Sciences, Czech Academy of Sciences, Lidická 971,
    602 00 Brno, Czech Republic; lukas.kala@ibot.cas.cz
3   Department of Environmental Studies, Faculty pf Social Studies, Masaryk University, Joštova 218/10,
    602 00 Brno, Czech Republic; tomaschabada@mail.muni.cz
\*   Correspondence: jana.krcmarova@yahoo.com

**Abstract:** Czech agriculture is dealing with the consequences of climate change. Agroforestry cultures are being discursively reintroduced for better adaptability and resilience, with the first practical explorations seen in the field. Scholars have been working with farmers and regional stakeholders to establish a baseline for making agroforestry policy viable and sustainable. In a research effort that lasted three years, a large group of Czech farmers was interviewed via questionnaire surveys, standardized focus groups and in-depth personal interviews regarding their knowledge of agroforestry systems, their willingness to participate in these systems, and their concerns and expectations therewith. The information obtained helped the researchers gain better understanding of issues related to implementation of these systems. It was found that although trees are present on Czech farms and farmers appreciate their aesthetic and ecological landscape functions, knowledge about possible local synergies with crops and animals is lacking. This local knowledge gap, together with lack of market opportunities for the output of agroforestry systems and undeveloped administrative processes, have been identified as the greatest obstacles to the establishment of agroforestry systems. The researchers argue that the discovered cognitive and technological "lock-in" of the farmers may represent a risk to climate change adaptability and resilience. For the development of complex and localised land use (e.g., agroforestry) in such a context, the researchers suggest participative on-farm research, which would broaden the local knowledge base related to ecology and entrepreneurship.

**Keywords:** agroforestry policy; participatory approach; local knowledge; climate change

## 1. Introduction

Industrial agriculture with high energy inputs and less manual labor is dominant in the Czech Republic. Production is focused on a limited number of crops, with the average farm area being 133 ha in 2018, the highest in the EU [1]. Machines and chemistry do most of the work. The farmers operate in compliance with the national legal system, take advantage of subsidies provided by the European Union (Maastricht, The Netherlands) through the Common Agricultural Policy, and sell their goods on the global commodity market.

In regard to knowledge production in the field of agriculture in Czechia, the Western scientific tradition dominates [2–5]. At the end of the 19th century a system of professional and higher education institutions was founded, which helped with the implementation of the industrial agricultural model throughout the country, a process well described in Czech lands [6]. The traditional agricultural system of local plant varieties, practices and schedules based mostly on organic fertilizers and human and animal labor has been transformed greatly since then. Since the second half of the 19th century the biocultural diversity of Czech farmland has fallen in a similar manner as in the rest of Europe. The

communist regime further eroded the role of oral tradition and the transmission of local agricultural knowledge within families.

There are several local fellowships and projects dedicated to guarding local agricultural knowledge (including landraces, instruments and products). However, the weakened knowledge base is generally not causing concern at higher policy levels. The connection between local knowledge diversity and the resilience of farmlands is rarely recognized. Even at the European level, the plurality of ecological knowledge systems was recognized only recently as offering sustainability and rural development opportunities [7].

At the biological level, industrial agriculture has similar negative externalities in Czechia as in other countries. Soil and water pollution and eutrophication, erosion, mechanical soil degradation, and a fall in biodiversity are some of the commonly seen by-products of conventional agriculture [8,9]. Recently the conservation or establishment of less intensive agroecosystems has been seen as a need to meet EU biodiversity goals [10,11].

In Europe as in the USA, experts are beginning to worry that such socio-ecological agricultural systems will have problems adapting to climate change. The Czech Republic is already facing the consequences of climate change in the form of reduced precipitation and decreased groundwater levels, precipitation pattern changes, floods, and droughts. In intensively cultivated regions, this can lead to soil desertification and pollution and to more frequent occurrences of pests and diseases in crops. The recent unprecedented *Microtusarvalis* population boom was connected with mild winters, no-tillage soil preparation and decreased predator fauna in farmland.

To achieve resilience and healthy socio-ecological systems, scientific knowledge must stay connected with societal and political processes [12]. The science of sustainability—a solution-oriented interdisciplinary field—suggests improvement of the scientific research process by inclusion of key stakeholders. This is considered as a prerequisite for building vital sustainable policies [13]. In solutions related to climate change, neither hard technological solutions nor soft social innovations are considered to be sufficient and a systemic approach concentrating on ecology, economy and society is needed [14]. As many problems have local or regional impacts, the regions start to play a vital role in the governance of mitigation and adaptation [15].

### 1.1. Technocratic Governance in Agriculture in Socialist Czechoslovakia

When discussing the industrialization of agriculture in relation to traditional/local and modern/expert agricultural knowledge, it is useful to relate it to matters of policy creation and governance. Contemporary historians argue that, in post-war socialist Czechoslovakia, technocratic governance was a fundamental component of state policies. Between the end of World War II and the "normalization" (consolidation) era following the 1968 Warsaw Pact invasion of Czechoslovakia, a rising trend of technocratic governance can be traced [16].

The transition seen in the field of agriculture in the second half of the 20th century in Czechoslovakia can be summarized as an expert-led conversion of agriculture into an industrial sector based on a combination of profit-oriented large-scale production run on business principles and the efficient organization of labor. Economic policies in socialist Czechoslovakia related to agriculture and food security (among other things) were firmly rooted in centralized rational planning, which was performed at several levels of the hierarchy and included management and control at all levels. This approach was meant to enable reliable forecasting of economic results and to ensure productivity and efficacy. Data collection was performed from the bottom to the top of the "pyramid"; planning was done at the top level and instructions were subsequently communicated in the top-down direction to individual agricultural enterprises. Let us briefly outline the historical developments in agricultural policies and practices in the post-war period and show the varying measures of state intervention in the management of agriculture.

In the first few years following the end of World War II, the Communist Party of Czechoslovakia effectively gained control of the management of agriculture. This brought a limitation of the pre-existing capitalist principles of the inter-war republic, (partially forced)

collectivization of small and mid-sized farms into agricultural cooperatives (completed in the late 1960s), and the implementation of government-level central planning. With this, a large step was taken towards the eradication of the tradition of small-scale agricultural production. The process which had been set into motion can be described as a "fast decline of traditional agriculture based on various forms of land tenure and time-tested production procedures (and) the purposive elimination of natural relations in agriculture", which had been developing since the mid-1900s [17].

In the 1950s, it was determined that the key to boosting underdeveloped agricultural production in Czechoslovakia was the introduction of mass-production technology, mechanization and the use of chemicals. Central planning had very high aims and burdened the new agricultural collectives and state enterprises with directive indicators of performance in production. The 1960s, however, started to see a general air of progressive social and economic development. Agricultural economic reforms of the early 1960s enabled the loosening of rigid indicators, and allowed for economic involvement, meaning that collectives were now able to make their own financial decisions about production (e.g., selling surplus production, increasing members' social security). Giving the collectives and state enterprises a degree of independence in management and decision-making of their affairs also resulted in better performance in production. Moreover, in the second half of the 1960s, in a general atmosphere of political liberalization leading up to 1968 (the "Prague Spring"), collectives began to form associations, the idea of self-government in agriculture was recalled and some attempts at the renewal of traditional agricultural cooperatives were made, bringing back the pre- and inter-war tradition of a variety of agricultural associations, albeit for a short period [17].

In the late 1960s, experts began considering interconnecting agriculture with the food-processing industry so it could be managed as a single industrial sector. These ideas resulted in the formation of a so-called "agriculture and food-processing complex" designed to harmonize the management of all sectors involved in food production by the government. Its main principles included increasing production while cutting costs and human workload. The plans created were supposed to take into account local and seasonal specifics, and the motivation of individual enterprises by supporting their profit interests [16]. With this, rigid central planning began to loosen in favor of liberalization of the economy.

The liberalization of society and agriculture, however, was suppressed by changes in the political representation following the 1968 Warsaw Pact invasion, termed "normalization" (consolidation). Directive central planning was reinstated, and scientific management of agriculture was promoted. Large enterprises combining agricultural production and food processing were established as a consequence of the idea of concentrating agricultural production which gained traction in the mid-1970s, as well as specialized enterprises [18], with the aim of reducing human workload by way of automatization, effective management and production planning [19]. However, these large organizational units proved very difficult to manage [17].

These modernization efforts, however, had significant downsides. Traditional agricultural knowledge and intuition were replaced with expert calculations and exact plans based on contemporary science and technology [16]; moreover, the goal of food security was only attained at the cost of long-term damage to the environment and irreversible landscape interventions [18] and involved deterioration of soil quality and food quality due to the use of agricultural chemicals.

Historians argue that socialist organization and management of agriculture, which was in place until the end of the socialist era in 1989, necessarily resulted in stagnation, the reasons being, among others, the obligation to follow rigid plans, limited freedom in decision-making, and resulting lack of involvement of farmers and workers in collectives and state enterprises [17]. The view of private farmers in the 1990s on expert knowledge and the role of public authorities in landscape management, characteristic of the era, is epitomized in a quote from a 1990s survey: "I do the work of ten clerks and environmental-

ists employed by the state. Not only do I know as well as they do what work needs to be done, but I can also do it with my own hands" [20].

### 1.2. Participatory Governance and Policy

The idea that decision-making is the sole responsibility of public authorities and experts has been losing ground in several ways since the transition following the year 1989, due to the (re)emergence of a civil society which demands public participation in policy shaping, the weakening conviction that science and technology are able to gain a perfect understanding of the world and society and give reliable expert advice, undermined confidence in the decisions and instructions of the technocratic elite and bureaucrats, etc. Leaving policy-making in the hands of unelected experts is viewed as a deficiency of democracy [21].

Within the international context, participatory policy-shaping has been defined and the process explained in a range of documents [22,23]. Article 7 of the Aarhus Convention clearly states that the public is to be invited to participate in the process of planning pertaining to the environment. The inclusion of the public in decision-making processes is guaranteed by the European Landscape Convention, ratified by the Czech Republic in 2004.

Relevant to the context of introducing agroforestry in Czechia, the benefits of public (i.e., chiefly farmers') participation in development of policies include better public policy, based on new sources of information gained from citizens, and more effective implementation, resulting from greater voluntary compliance, as the needs and expectations of citizens are taken into consideration [23]. However, experts show that this may not always be easy (participatory governance may have to face skepticism, apathy or rejection in post-communist countries) and suggest paying attention to cultural contingency in participatory governance [24].

### 1.3. Agroforestry Benefits and the Implementation Gap

The reality of climate change calls for transformation of all production sectors including agriculture. From the mitigation measures at hand in primary production, agroforestry has the greatest potential in this regard. Permanent or semi-permanent tree cultures function as a great carbon sink, reducing the concentration of $CO_2$ [25]. Tree vegetation also directly cools the atmosphere and land cover (e.g., [26,27]).

There are other benefits of mixing trees with agriculture, such as the enhancement of water filtration and infiltration; water retention; soil fertility, nutrient balance and biota diversity; microclimate for crops and livestock; habitat opportunity for wildlife, and others. In the era of post-industrial agriculture in Europe, the agricultural landscape is supposed to fulfil other functions besides food production for the population. The aesthetic character of trees interspersed in the farmland is also valued [28]. In the USA, agroforestry is praised for advancing all five core environmental concerns of so-called regenerative agriculture: soil fertility and health, water quality, biodiversity, ecosystem health, and carbon sequestration [29].

Scholars have for several decades suggested agroforestry as a possible win-win-win in land use (bringing environmental, social, and economic benefits), because it can provide food and fiber, maintain habitat for threatened species, and provide work opportunities and a healthy landscape for humans [30]. The European Commission has also suggested agroforestry as a mechanism for adaptation to climate change and for its mitigation [31]. This policy is based on the FAO climate smart agriculture concept and the transition towards ecological intensification. The systems envisaged in the latter, e.g., agroforestry, should be highly productive, economically viable, environmentally sound and based on the principle of equity and social justice. In general, healthy agroecosystems are seen as significant contributors to food security and the socio-economic viability of the countryside.

While mixing trees with agriculture has a possibly Neolithic origin in temperate zones [32,33], modern agroforestry has been more commonly studied and practiced in the

tropics. In these zones the research to mitigate the effects of deforestation, resource degradation and poverty already led to agroforestry measurement in the 1980s [34]. The research and development institution ICRAF (World Agroforestry Institute) founded in 1977 also concentrated primarily on tropical ecosystems. Although agroforestry in temperate regions is gaining more attention as we speak, there is still lack of knowledge about the functioning of modern agroforestry systems at the European level. The project AFINET (AgroForestry Innovation NETworks) funded by the European Commission through the Horizon 2020 program addressed this and filled in some gaps. With the use of an agroforestry innovation network, information is collected and shared among partner countries participating in the program, forming the basis for innovation. It is acknowledged that agroforestry systems function differently in different contexts, climates and agricultural sectors. From multifunctional hedgerows close to nature to introduced poplar bioenergy crops, agroforestry has many forms and can perform many functions.

In central Europe, mixing trees with agriculture is considered an ancient land use system. This is natural—the local vegetation climax is a forest, and treeless regions are either cultural or rare (mountain tops, specific edaphic conditions), so trees have always been actively removed from fields and pastures. In Czechia, the coexistence of agriculture and trees was considered normal at least until the middle of the 19th century—at the time, mixed cultures were still a common type of land use with a special taxing category [35]. However, during the industrialization of agriculture that started in the late 19th century and intensified especially after World War II, these cultures were outcompeted and forgotten [36]. They were erased both from the agricultural theory taught in schools and higher education institutions and from landscape classification [37]. Nowadays, only remnants can be found in areas which were not suitable for intensive agriculture or forestry [38].

Agroforestry in the Czech landscape is currently undergoing a revival. In 2014, the Czech Society for Agroforestry was founded (Českýspolek pro agrolesnictví, www.csal.cz (accessed on 3 December 2020)) via an initiative supported mainly by the Faculty of Tropical AgriScience at the Czech University of Life Sciences Prague, the Mendel University in Brno and the VÚKOZ research institute which has been researching poplar bioenergy crops for some time. Today, this agroforestry society includes a diverse spectrum of members from scholars to farmers to politicians. The society has done a large amount of educational work at the national level, holding its first seminars for farmers and initiating its first projects; currently it is preparing the first book about Czech agroforestry. Members of the society have also been lobbying the Ministry of Agriculture for the recognition of agroforestry in the design of new national CAP policy measures. The year 2020 saw an exhibition dedicated to agroforestry in the National Museum of Agriculture, Prague, Czech Republic, which also attracted the attention of the Ministry of Agriculture.

Even though the topic is receiving more and more attention in scholarly and political public arenas, there is still an implementation gap. Agroforestry is practiced only by a small number of farmers. The drive to increase the presence of trees in the agricultural landscape is currently coming more from scholars, from various backgrounds and political representation, which is supposed to address current issues of climate change and other environmental concerns.

Exposing and examining the concerns and doubts farmers may have towards implementing agroforestry systems on their farmland is a key step for preparing proper funding schemes, anticipating potential issues, and tailoringthe adjusted measures and conditions in order to help increase compliance and prevent rejection of the new agroforestry policy.

## 2. Materials and Methods

Perception and knowledge of Czech farmers regarding keeping trees on farmland and in agroforestry systems was gathered in 2018 and 2019. Two regions (out of 16) were chosen for analysis—Central Bohemia and South Moravia, the longest inhabited, warmest and most fertile regions in the Czech lands in which Czech agriculture started. Today their landscape is to a great degree transformed by conventional industrial agriculture. They

are also the two regions facing the most profound issues concerned with drought—one possible climate change consequence. Before the research, the regional authorities were contacted and expressed their willingness to use the results in further work.

Multiple methods were used including focus group discussions, wider questionnaire surveys and qualitative in-depth interviews. The results were compared in 2020 so the information could be synthesized and presented in this article. Qualitative methods were used to gain insight into farmers' opinions and knowledge about agroforestry, similarly to [39–41].

Focus groups were organized in the two regions in 2018 and 2019. They followed seminars led by experts, who focused on highlighting the socio-economic and environmental benefits of agroforestry (a similar methodology has been used by [42]). Focus groups were led by trained moderators who kept to a semi- structured interview script designed to cover research questions. The respondents (in total 113) were mostly local farmers without any experience with agroforestry and the majority were farming less than 50 hectares of land. The audio-visual records from the research sessions were selectively transcribed using the F4 program with time coding. Transcripts together with field notes were exported to the Atlas.ti software. The qualitative data was analysed for clustering of words and themes. The arising clusters identified keywords/key themes for further investigation.

Findings from the qualitative focus group research were used to design the subsequent questionnaire survey. The questionnaire was prepared in cooperation with stakeholders and was distributed among Czech farmers in the form of on-line and off-line data collection. The questionnaire was published in the Selská revue journal published by the Association of Private Farming of the Czech Republic and also distributed by post to a random sample of private farmers and farming companies in the whole of the Czech Republic. The questionnaire collected data on the number of woody plants on the farm and reasons for keeping them, the willingness to found an agroforestry system on owned or rented land, preferences regarding silvo-pastoral or silvo-arable systems, and expectations and concerns regarding the founding of agroforestry systems. Participants were asked to respond on a 5-point Likert scale ranging from strong disagreement to strong agreement. The researchers collected 151 completed questionnaires, 92 from Central Bohemia and 59 from South Moravia. The group included 90 private farmers, 47 agricultural companies and 14 cooperatives from these two regions, but only 34 subjects identified themselves as organic farmers.

In-depth interviews were conducted with 30 farmers from the two regions in question (Central Bohemia and South Moravia). Twenty of them are private subjects with smaller farmed areas and 10 manage farmed areas greater than 200 hectares and/or are managed as enterprises. Farmers concentrating on both crop and animal production were included, as well as one beekeeper. Through two personal visits and one phone call to each of the farms, the theme of "trees on farms" was investigated. During the first visit, a free-listing focused on the theme of "trees on agricultural land, drought and erosion" was used. This type of interview allowed the researchers to delineate the cognitive domain of the subject—the range of concepts, perceptions, and notions which the farmer connects to the given theme. During the second visit the subjects sat through a semi-structured interview concentrating on local ecological knowledge using the adjusted methodology. The framework included questions about ecological relationships in the farmers' agroecosystem with special emphasis on the interactions between trees and crops, farm animals, soil biota, wildlife and local, regional and global climate. The interview also included a discussion of the legislative, economic and cultural barriers and concerns regarding productive agroforestry and keeping trees on farms. At the end of the second research interview the researchers employed the photo-elicitation method using pictures of European silvo-arable and silvo-pastoral agroforestry systems from France, Italy, Spain, Switzerland, Germany, Hungary, and Romania. With the introduction of functioning systems, farmers discussed the potential advantages and barriers to implementing agroforestry systems in the Czech Republic in general, as well as more specific expectations and concerns stemming from their own experience with implementing agricultural programs as such. The resulting 200 h of audio recordings from

the interviews were anonymized and transcribed in the F4 transcription program with time coding. Selected parts were analysed in the MAXQDA 2018 software using thematic analysis. The primary structure of themes was based on the researchers' theoretical and conceptual background; additional themes were coded as new as significant ones emerged, until thematic saturation was achieved [43].

In total, the researchers collected and analysed the opinions of 113participants in focus groups, 30in-depth interviews and 151questionnaires from the two key regions most affected by climate change. The research process is presented in brief form in Table 1.

**Table 1.** Summarization of the research process.

| Year | Regions | Methods | Data Characteristic | N of Fespondents (Farmers or Agricultural Enterprise Representatives) | Common Research Objectives | Data Treatment and Analysis Method |
|---|---|---|---|---|---|---|
| 2018 and 2019 | South Moravia | Focus groups | Semi-structured questions | 57 | Trees on farmland (their presence, landscape context, reasons for keeping them, ecological knowledge about relationships between trees and agroecosystems); agroforestry systems (knowledge, obstacles, opportunities); with the in-depth interviews data were gathered also about farmers opinions regarding the climate change | Transcription of audio records taken at focus groups and interviews (with F4 time coding software) and of filled-in surveys; qualitative analysis with atlas.ti and MaxQDA software |
| | | In-depth interviews | Unstructured and semi-structured questions | 15 | | |
| | | Survey | Structured questions | 59 | | |
| | Central Bohemia | Focus groups | Semi-structured questions | 56 | | |
| | | In-depth interviews | Unstructured and semi-structured questions | 15 | | |
| | | Survey | Structured questions | 92 | | |

## 3. Results

The three methods used (focus groups, questionnaire survey and in-depth interview with farmers of the two most agricultural regions in the Czech Republic) brought representative insight into the role given to trees, alleys and woodlots on farmland. With the exploration of the farmers' perception of trees, the gap in implementing agroforestry solutions was also addressed, since agroforestry per se is still very rare in the Czech Republic. The two regions in which the research took place would greatly benefit from the application of agroforestry practices as these regions are also the ones most prominently affected by climate change consequences—droughts, water erosion, soil degradation and pest outbreaks.

This chapter presents the synthesized results and information (gathered using the three aforementioned methods) about the farmers' knowledge and awareness of the ecological relationships between trees and the agroecosystem and their perception of agroforestry farming in contemporary economic and political systems.

### 3.1. Trees Have Always Been There

Keeping trees on the farmland is a widespread practice among the interviewees, although the median number of trees on farms in the two heavily ploughed regions is low (five trees). The presence of trees on the farmland is largely not a result of active tree-planting, nor is it driven by their economic value. In most cases, the trees simply remained on the farmed areas in non-production or hard-to-reach places, sometimes forming sparse or dense alleys around the fields or woodlots on the margins.

Although due to the methods and objectives chosen we were not able to gather information about tree species on farms by frequency, through in-depth interviews we did note the trees noticed by farmers on their farms. They were (in alphabetical order): *Acer platanoides, Aesculushippocastanum, Alnusglutinosa, Carpinusbetulus, Crataegus* sp.,

*Fagussylvatica, Fraxinus excelsior, Juglansregia, Malusdomestica, Morus alba, M. nigra, Piceaabies, Pinussylvestris, P. nigra, Populus* spp., *Prunuspersica, P. avium, P. spinosa, P. cerasus, P. armeniaca, P. domestica, Pyruscommunis, Quercusrobur, Q. petraea, Q. rubra, Rosa spinoza, Robiniapseudoacacia, Salix* spp., *Sambucusnigra, Sorbus* spp., *Symphoricarpos albus, Tiliacordata, T. platyphyllos, T. x vulgaris.*

Farmers thus do have experience with trees interacting with crops or animals. In their experience, trees pose technical limitations for ploughing (difficult access for machinery, damage to machinery caused by hitting tree branches, etc.). Many farmers notice competition for water and sunlight between their crops and the trees. On one hand, they agree that trees retain water; on the other hand, they think that when there is enough precipitation, the overall balance is negative for crops because the crown of a grown tree forms an "umbrella" and more water is drained from lower ground levels due to increased evapotranspiration.

Although farmers do not see many ecological benefits in the trees growing on their farmland, they keep the existing balks with trees and woodlots where they are. The main reason expressed in the questionnaire survey was the preservation of the aesthetic of the landscape (cf. [43]); many farmers also value the anti-erosion function of the tree balks and to a lesser extent the potential contribution of tree growth to the improvement of the quality of the soil and microclimate (see Figure 1). The synergic effect between trees and agriculture was more appreciated by some pastoral farmers, who for example emphasized that trees serve as a natural shelter for cattle in pastures (cf. [44]), that nectar-bearing trees are important for bees, and that trees help create a moist microclimate. In many cases, the subjects simply admitted during the in-depth interviews that they let the trees be, without deeper thought. Some claim they simply do not have the capacity to either eradicate them or cultivate them further and they treat them as simple landmarks.

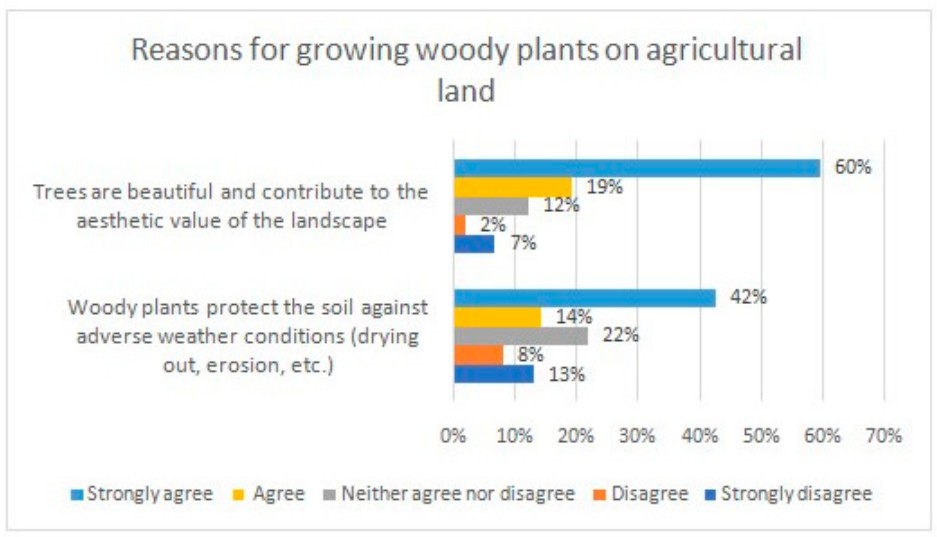

**Figure 1.** Farmers' perception of woody plants as indicated in the questionnaire survey.

Some of the interviewees actively plant trees in their landscaping efforts, aiming at aesthetic value and diversity of the landscape. They also plant fruit-bearing trees to restore orchards, but this is a lower-priority activity because of reduced market opportunities and agricultural laboriousness.

In general, surveyed farmers do not agree on the question of whether it is necessary to have trees on agricultural land. Leaving aside answers that suggest a lack of interest in the topic, the respondents' attitudes range from a clear "no", based mainly on fears of suppressing the productive function of the soil, through calculations comparing soil yield and the environmental benefits of the trees, to positive attitudes motivated by seeking

ways to cope with the problems posed by drought and other changing climatic events (see Figure 2).

**Figure 2.** Expectations of farmers regarding growing woody plants on agricultural land found in the questionnaire survey.

### 3.2. Agroforestry Is an Unknown Territory

Since agroforestry has not yet been widely implemented in the Czech Republic, the local farmers have very little experience and knowledge of its ecological, technical and administrative aspects.

The photo-elicitation approach with photos of viable and functioning agroforestry systems across Europe presented during the in-depth interviews and during the expert seminar introducing the focus group discussions generated some ideas about the possible utilization of agroforestry in the Czech Republic. When presented with examples from abroad and with the general principles, the subjects could see the theoretical potential benefits to production, ecological function and synergistic effects.

Alley cropping systems were seen as possibly aesthetically pleasing; however, as the farmers have the experience that trees along their fields compete with the crops, they questioned whether such systems could be simply transferred into the Czech environment (if yes, the farmers would need to know which plant varieties should be used under such conditions). Regarding tree species, the farmers are accustomed to having fruit trees around the field and would not be opposed to having more of them—if there was market for the produce and enough labor (cf. [40]). As for trees with valuable wood, the farmers feel disoriented in the market and also worry about the amount of time needed for the investment to pay off, while giving up some producing land now.

When discussing animal husbandry systems, one of the most unproblematic systems were hens kept in orchards, which is a preserved family practice surviving up to the present time in some areas. The farmers agreed that free movement and diversity of food improves the quality of chicken eggs and the feces improves the quality of the soil. Similar benefits were observed with sheep and cattle in orchards; however, the farmers cited obstacles in the contemporary prevalence of low stemmed tree orchards and insufficient market opportunities for local fruit, wool, and mutton. Agroforestry systems with pigs in forests or in bio-fuel tree plantations were rejected due to the legislative obligation to keep pigs mostly indoors due to the risk of African swine fever virus disease and the overpopulation of wild boar in Czech lands.

There is a frequent fear expressed by the farmers that it is impossible to predict the functioning of agroforestry systems in the local context as there is not enough experience of it in the Czech Republic.

An important and recurring condition for founding agroforestry systems is that it would not be necessary to change the technical equipment of the farm in order to maintain the trees, or to significantly interfere with the other production sections of the land (cf. [43]). The farmers who participated in the research (and were not totally opposed to agroforestry) also requested a complex expert evidence-based information campaign to complement any arising policy. In the questionnaire survey they preferred education in the form of educational field trips, educational seminars and a paper-based handbook (see Figure 3). The farmers had similar requests during the in-depth interviews—they wanted to consult experts about the specifics of planting, harvesting, the types of trees to be planted, the design of the system, the crop combination, where to obtain seedling material, what products to focus on and where to find a market for them. Regarding the new policy, the majority of the farmers in the in-depth interviews feared that the rules will be too rigid, with insufficient room to design systems flexibly in accordance with local specificities and farming regimes. For example, they would appreciate the opportunity to grow their own seedlings, rather than buying larger quantities from other suppliers.

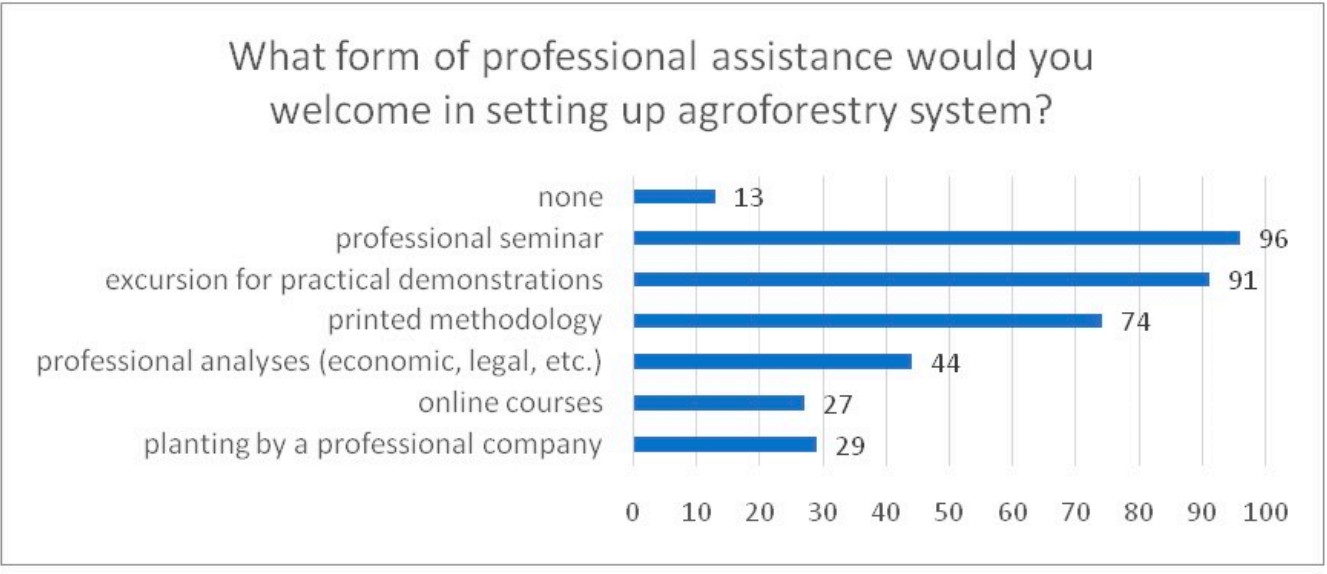

**Figure 3.** Forms of help farmers prefer in the questionnaire survey.

At present, although trees are welcome, their more widespread placement in the landscape connected with bigger investments and yield expectations is perceived, rather, as a source of risk. Most of the farmers frame their thinking with production as the primary function of agricultural land. Investing in agroforestry under the current conditions is seen as a possible existential threat, stemming from the need for technologies and know-how, with possibly weak return on investment. Farmers often assume laboriousness of farming in agroforestry systems, and the increased workload is a further concern, as farmers are already facing labor shortages (cf. [40]). Their concerns are often related to the belief that the demand in the Czech market for the products of agroforestry systems (fruit, wood, wool, meat, etc.) is low, or from their own lack of awareness of the possibilities of selling these commodities.

Farmers also have concerns regarding the administrative burden of the bureaucratic process of the introduction of agroforestry practices (cf. [43]). Ownership and tenancy legislation, as well as the system of subsidies, are widely perceived as significant obstacles. It is complicated for farmers farming on leased land to plant trees there. Other farmers see the threat of losing direct payment subsidies, which are still bound to tree-less areas.

There is also a widely shared view that planting trees requires a long-term investment, in addition to the uncertainty as to whether seedlings will root and grow well—the farmers surmise that the positive effects of trees in the landscape will not be seen for several years or decades, if at all (see Figure 4).

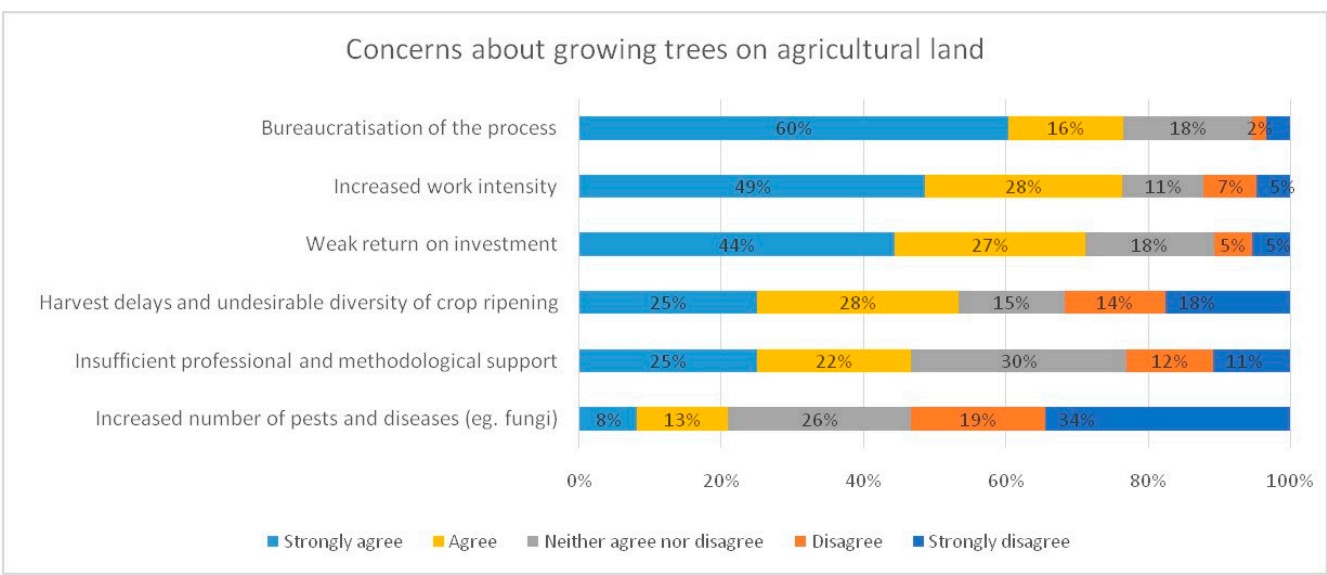

**Figure 4.** Concerns of farmers regarding growing trees on agricultural land found in the questionnaire survey.

As for the willingness to start an agroforestry system on a part of farmland, there is a marked difference related to the legal ownership status of the land. As there is a legal obligation not to alter the key features of rented land, the farmers would mostly agree to found such a system only on their own land. In the questionnaire survey, organic farmers were more open to adopting agroforestry farming.

Those farmers who can see their land as more than a production unit do, for example, notice that wildlife diversity has dropped since the time of their parents or their own childhood, which they attribute both to the chemicals and machinery used today but also to the loss of shelter provided by permanent (tree) landscape elements. Many of them note that wildlife is a part of farmland and should be able to find some shelter. However, all subjects primarily consider the economic sustainability of their farm.

### 3.3. Climate Change Is Not a Sufficient Driver Yet

The introduction of agroforestry is suggested by scholars and policy makers as one of the climate change adaptation measures. Part of the in-depth interviews focused on the importance that interviewees ascribe to vegetation and tree cover and also agroforestry within the context of the causes of and solutions to these changes.

The majority of farmers do not dispute climate change per se. The farmers' beliefs regarding the causes of global climate change are mostly in line with established scientific understanding of human-induced climate change. Some former sceptics admit they were convinced once the expert predictions became locally-manifested reality. Specific causes connected by the farmers to climate change were the use of fossil fuels, industrial agriculture (specifically the use of industrial fertilizers), and disregard for conservation of the landscape. A few interviewees remain skeptical and maintain that local events such as droughts, changes in precipitation patterns, etc., are instead caused by natural cycles. There are also varying views on the contribution of industrial agriculture to climate change.

The farmers' understanding is marked by a disconnection of global-scale climate change from its local manifestations. In other words, some farmers are not convinced that it is climate change that causes the various on-the-ground issues they face. Instead, the subjects see the roots of these issues in local factors: for example, droughts and rising

temperatures are attributed to the proximity of a large airport and air lane, and of a urban heat island (in the case of the Central Bohemian region) and vast fields (in the case of the South Moravian region).

The most profound manifestation of climate change that most farmers notice in these two regions of the Czech Republic are the changes in precipitation patterns and distribution. Most note that the distribution of precipitation throughout the year is less even, with rainfall more concentrated and heavier at the same time, and with storm intensities increasing. In drier years, spring rains are delayed (for example to late May), which impedes the growth of spring crops and also results in lower crop quality. A significant portion of the year's precipitation volume falls in late summer—which is "too late for crop farmers to be of any use". The heavier rainfall instead hinders harvesting because the crops are wet. Some even go so far as to say that a whole season (spring) was skipped for several consecutive years. Farmers also see decreased groundwater levels (observed in wells), drying ponds and streams and decreasing soil moisture. The water deficit is apparent not only on arable land, but also in forests where farmers observe trees growing at a slower pace or dying.

A crucial area in which farmers observe the impacts of climate-change-related processes is the decline in yield. This is primarily an issue with spring crops. As stated above, farmers observe that the spring rains arrive too late for crops to grow in adequate volume. Some have noted a decrease in yield of grains and oil plants of up to 40%. Spring drought impacts crop quality as well, forcing farmers, for example, to sell grains as fodder rather than as human foodstuff and to lose part of their profit. Low and low-quality yield is moreover attributed to changes in the seasonal cycle. Apart from water shortages, spring also gets shorter as high temperatures arrive early in the year. Farmers note that plants have too little time to regenerate and grow sprouts and that the higher temperatures cause premature blooming.

Yield is further threatened by pest population booms. Farmers see more frequent pest outbreaks, for instance in grains and oil crops. This is seen as a result of higher temperatures and necessitates increased use of insecticides. On the other hand, the warm and dry conditions reduce the occurrence of fungal diseases. Nevertheless, farmers choose to rather "be safe than sorry" and apply fungicides pre-emptively. The usage of disc harrowing instead of deep ploughing to maintain moisture in the soil, together with mild winters, also caused a population boom of voles, which in some fields in South Moravia caused yield losses of up to 90% in 2018 and 2019.

Farmers also notice changes in the behaviour of wildlife—for example, the occurrence of migratory birds, so far never seen nesting, or overwintering of normally migrating birds.

Practical on-the-ground responses to the water scarcity and soil erosion implemented by the interviewees include contour ploughing, lowering the intensity of grazing, disc harrowing, ploughing in the vegetation remnants after harvest, ploughing grass catch crops, and applying green manure or animal manure. Rainwater harvesting systems have been cited as a very beneficial measure for the collection of water, along with the adaption of widespread existing drainage systems from the 20th century. Farmers also adopt new technologies and techniques. These include introducing drought-resistant plant varieties, varieties adapted to warmer conditions, and early-maturing varieties. Some farmers also experiment with non-traditional crops, e.g., sorghum and foxtail millet. To prevent low or low-quality yield, some mention sowing corn deeper into the ground and applying herbicides immediately after sowing to get ahead of plant competition for water.

In regard to the understanding of the role of tree cover in climate change, many of the farmers agree that the presence of wood and woodlots cools the landscape. They also understand the benefits of keeping trees on pastures as a microclimate improvement. They also value tree alleys as windbreaks and appreciate them, along with grassy strips and balks, for their protective role against soil erosion caused by water and wind (which is becoming more serious due to the changes in climate). However, they do not give them much value in the context of the overall local water balance and they do not mention the role of trees in carbon sequestration at all.

## 4. Discussion

The tradition of using trees in agriculture in Czech lands was broken during the industrialization period and nearly completely forgotten since. The renewal of agroforestry in the country should be perceived as a complex developmental issue. The research has shown several interconnected points which may hinder the implementation of this approach to land use among Czech farmers. The most important of these points are the lack or loss of experience and knowledge, the cognitive and technological lock-in, the contemporary identity of farmers, their attitude to climate change and their understanding thereof.

In European history, ways of combining trees with crop cultivation or animal husbandry were naturally variable and locally adapted, thanks to both biological and cultural diversity. This means there is not just a single correct way of reintroducing agroforestry, but rather a myriad of locally possible options. The authors of this article worked with the concept of local knowledge, with the expectation of a certain level of experience coming from direct contact with agricultural ecosystems.

In the two regions that were studied, the trees or bushes had simply remained in their place on farmlands from earlier times. The contemporary woodlots and alleys are small remnants of all the elements that were present in the pre-war agriculture landscape. In the pre-industrial or early industrial farming period the farmland was finely divided into small plots by balks with trees and bushes and hedgerows, most probably used for animal grazing or/and fruit picking [35]. However, most of these landscape elements were removed in the communist era after World War II, when the privately-owned plots were united into great fields and heavy machinery and chemicals started to be utilized on a large geographical scale [45].

Farmers do not seem to either view trees as a nuisance or consider them very important from the economic point of view, which for most of them is the primary factor considered in the decision-making process regarding land use. They do find the tree elements aesthetically pleasing, whether they are present as part of an agroforestry system or for no specific purpose. However, the results of the research have shown that the farmers lack the needed knowledge of how to successfully incorporate trees in the agroforestry systems in their region and landscape type. As in [46], Czech farmers do not possess enough experience and thus knowledge regarding synergies between trees and crops or animals. They mostly expect competition between crops and trees and fear further losses of water from the soil. This cognitive lock-in manifests itself in prejudices and stereotypes concerning the cultivation of woody plants on agricultural land.

Even though there are many studies from abroad presenting various synergies in alley cropping or silvo-pastoral systems, the synergistic effect between trees, crops and animals is not fully acknowledged in the Czech Republic because such systems are still very rare in the country. Czech farmers view the simple transfer of agroforestry experience from Atlantic or Mediterranean or mountainous biogeographic regions with suspicion. Agroforestry is such a complex and locally adapted approach to land use that its reintroduction requires much wider participation from farmers, which would form a base for proper in situ research. However, the majority of farmers who would consider implementing agroforestry, would instead expect tailor-made solutions delivered in a top-down way. Only a few said they would appreciate more flexibility and freedom given by the government. Most do not perceive themselves as knowledgeable and sufficiently skilled to do agroforestry or innovate on their own.

From the socio-cultural view, the low interest in and knowledge of agroforestry in Czech farmers can be a historical maladaptation. As discussed in the introduction, agriculture has since the industrial revolution concentrated on yields without considering the rest of the landscape. Moreover, for more than a century the production and reproduction of agricultural knowledge has been primarily taking place in centralized education institutions supporting standardized industry-oriented production procedures. The topic of agroforestry has been ignored by agricultural science since the 19th century [37]. For even longer, agriculture has been separated from forestry and ecology. It is no wonder that

farmers do not feel knowledgeable and confident to work with trees. Here a remarkable difference was found between conventional and organic farmers, where the latter feel more skilled and inclined to work with trees.

The majority of the interviewees from the two most productive regions of the Czech Republic, however, seem to be 'locked in' a conventional style of agriculture (term first used by Louah et al. [46]). The most important lock-in identified in the cognitive space of the subjects lies in dependency on investment returns. This narrative was very frequent even among ecological farmers. The interviewed farmers show quite conventional approaches to increasing productivity. They seem to be locked in short-term solutions and risk being unsustainable practitioners. They rely on the application of synthetic fertilizers, pesticides and modern farm machinery. Under such conditions, it is not illogical that they are relatively conservative in their attitudes—they must not jeopardize the yields since they have families, employees, animals and loans.

The farmers expect production-oriented innovations, yet so far there is not sufficient know-how; even experts do not know exactly what to grow, where to grow it, in what combination, and under what regime to guarantee selling opportunities for a reasonable price in an affordable time frame. To a certain degree, rapidly growing bio-fuel woody plants form an exception; considerable developmental work in this area has been done by the VÚKOZ Institute, Prague, Czech Republic. Determining suitable tree species and crop combinations in various local conditions of specific landscape types, which would be profitable and marketable, is of crucial importance. So far, under current conditions, the farmers do not see ordinary trees helping their types of crop.

The role/identity of agro-forester simply requires different skills and knowledge—ecological, economic and technological. Furthermore, one needs a certain entrepreneurial spirit and economic flexibility to be able to open up and embrace a new role. It seems that Czech farmers are predominantly productivists [47]. They perceive the identity of environmental stewards or climate change solution-seekers as externally ascribed; however, the identity of somebody who bears responsibility for "feeding the country" is seen rather as internal. Many farmers deny the current societal perception of farmers as the main guardians of landscape, rather than as strictly agricultural producers. Many also do not perceive the implementation of agroforestry as a process of technological innovation contributing to sustainable intensification of Czech agriculture, but rather as a result of the efforts of environmental lobbyists and government officials to mitigate the effects of climate change by afforesting the landscape. They identify the declared benefits of agroforestry as a pretext for passing the responsibility for drought and biodiversity loss onto farmers themselves.

Czech farmers are well aware that they are entrepreneurs in a globalized and subsidized-agri-food system. A common lament during the in-depth interviews was that the prices of crops and meat from the average farm in the Czech Republic are dependent on the global commodity market, controlled by greater players—global retail chains, governments, supra-governmental bodies such as the European Union, and their relations. Since prices steer the farmers' decisions, the economic environment influences their decisions even more than climate change. They claim that ironically, in some cases the farm actually benefited from a dry year in the Czech Republic as the stock market for some crops allowed greater prices globally.

The farmers do not deny climate change, they tend to feel that it is anthropogenic, and they see changes on the ground (like the droughts and heatwaves in the period 2016–2019). They know that climate change risks cannot be separated from livelihood risks (cf. [48]). Because of this, they may be more prone to seeing possible threats to their farming [49]. For some, increased environmental risk awareness brings feelings of responsibility [50] and readiness to implement adaptation mechanisms. While the researchers found that farmers are looking to modify their practices, their actions are usually undertaken solely to prevent or counter the immediate effects of droughts and water scarcity, rather than being adaptive responses to climate change as such (cf. [51]). Therefore, they innovate and adapt under

immediate stress, while in the years with normal precipitation and temperature they tend to forget about climate change (similarly [52]). Consequently, they are not yet looking for long-term changes in their farming style or land us.

## 5. Conclusions

To broaden the stakeholders' participation in land use, policy-making means to give a voice and power to farmers. They, alongside policy-makers, scholars, environmentalists and others, are expected to enrich the knowledge base, and by doing so ensure that policies are viable and sustainable. Building an agroforestry policy "bottom-up" in the two Czech regions mostly affected by conventional agriculture and also recent droughts has shown both the promises and the limitations of this approach. The majority of the interviewed farmers do not know much about agroforestry and mostly they expect a top-down expert methodical guidance with a politically clear incentive and a system of subsidies if they are to participate in this approach to land use. Local knowledge of tree management has fallen into disuse and farmers today do not feel skilled or authorized to adopt the role of multifunctional agro-foresters on their own.

However, as the farmers in the studied regions are already dealing with changing precipitation patterns, water shortages, floods, erosion, low yield years, soil pollution and pest population booms, they put some thought into possible alternatives in case the climate change were to worsen. However, they are doing so within the limits of their business orientation and technological and knowledge base. When offered the possibility of agroforestry, they hesitate to accept it. From the global economic perspective, climate change is not enough of a driver of change yet for them to innovate on their own.

The researchers argue that the farmers may be cognitively and technologically "locked in" to unsustainable, heavy machinery-oriented, artificial fertilizer- and pesticide-based management (cf. [46]) based on their experience that the alternative measures of system performance are not able to compete. Therefore, agricultural policies should aim to show that there are other viable approaches than those in conventional agriculture. Cognitive and technological diversity has been proclaimed as a basis for possible adaptation measures in agriculture. It remains a question whether the radical innovation presented in the European Union's Common Agricultural Policy can occur in the regime of predominant conventional agriculture because it might be too locked into routine, established procedures.

Creativity and flexibility in farming strategies—or in land use in general—may be prerequisites for good adaptability to outer influences and unexpected changes including climate change. However, the results may be showing an alarming trend: the submission of alternative knowledge to prevailing conventional agriculture [53] together with limited willingness and entrepreneurial spirit among farmers operating in a narrow technological and economical manipulation space.

The support of on-farm research is a possible solution. There are good results with this in developing countries but also in Europe, where it has been studied in the context of implementation of complex innovative measures—sustainable farming systems. As Somers and Röling pointed out in 1993 [54], the interaction between scientific knowledge developed by experimental research stations and experimental knowledge developed on farms benefits the implementation process. Carrying out on-farm research with scientific standards and support from experimental research institutions can help establish a wider knowledge base. As the authors note, this can save some of the land and resource use alternatives, which might be lost if the policy was designed on a broader level. The diversity of agricultural knowledge stemming from such activities supports greater adaptability in these uncertain times. Another benefit is the improvement and development of the capacity of countryside communities, where social learning strengthens the farmers' creative and self-entrepreneurship abilities.

**Author Contributions:** Conceptualization, J.K. and L.K.; methodology, J.K. and L.K.; software, J.K., L.K., A.B. and T.C.; validation, J.K., L.K., A.B. and T.C.; formal analysis, J.K., L.K., A.B. and T.C.; investigation, J.K., L.K., A.B. and T.C.; resources, J.K., L.K., A.B.; data curation, J.K., L.K., A.B.; writing—original draft preparation J.K., L.K., A.B.; writing—review and editing, J.K., L.K., A.B. and T.C.; visualization, J.K., L.K. and T.C.; supervision, J.K. and L.K.; project administration, J.K. and L.K.; funding acquisition, J.K. and L.K. All authors have read and agreed to the published version of the manuscript.

**Funding:** This research was funded by Technology Agency of the Czech Republic, grant number TL01000298; The Institute of Botany of the Czech Academy of Sciences, a long-term research development project no. RVO 67985939, and Institute of Ethnology of the Czech Academy of Sciences, a long-term research development project no. RVO: 68378076.

**Institutional Review Board Statement:** Not applicable.

**Informed Consent Statement:** Informed consent was obtained from all subjects involved in the study.

**Data Availability Statement:** Not applicable.

**Conflicts of Interest:** The authors declare no conflict of interest. The funders had no role in the design of the study; in the collection, analyses, or interpretation of data; in the writing of the manuscript, or in the decision to publish the results.

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
