# Peer review of "Building Agroforestry Policy Bottom-Up: Knowledge of Czech Farmers on Trees in Farmland"

_land, doi:10.3390/land10030278_

Round 1

Reviewer 1 Report

Thanks for responding to all reviewer comments, the manuscript has substantively improved and now provides a good stock-take of current status of agroforestry in your country.

Reviewer 2 Report

The paper quality has been considerably improved. Providing the editor decides the style is correct (minor mistakes in sentences formulation can still be found) and presentation of the methodology sufficient for Land journal requirements, I would recommend to publish the manuscript.

This manuscript is a resubmission of an earlier submission. The following is a list of the peer review reports and author responses from that submission.

Round 1

Reviewer 1 Report

The manuscript describes opinions held among Czech farmers about a potentially increased presence of trees in agricultural landscapes -- which the authors see as desirable. Current farmers have little connection with traditional agroecological knowledge of their area and are (justifiedly?) skeptical of the 'new' agroforestry concepts and ideas. 

Against the specific political history of the Czech Republic, it isn't quite clear where the drive to explore 'agroforestry' originated and which and whose problems it may solve. Is it another imported fad?

Although the 

Title: you can't have a full stop in a title. Could it be: "Building agroforestry policy bottom-up: knowledge of Czech farmers on trees in farmland."

Line 11/12 "the agroforestry cultures are reintroduced" is ambiguous -- is it a discourse getting traction or actual change in tree cover?

Line 31 A statement like "the monoculture fields are on average hundreds of hectares" needs a reference.
Line 31 Does 'chemistry' do any work?
Line 36 References will have to shift to numbering system
Line 37 A reference would be nice here
Line 50 Sweeping statements like thus need references
Line 76 A time-line diagram might help here for eeaders not versed in political history
Line 122 Do you mean that the interest in and drive for agroforestry starts from 'public' participation, not from farmers or 'experts'?
Line 134 A reference on the cooling would be appropriate, e.g. Jan Pokorny's work
Line 152/153 References needed.
Line 218 legal subjectivity?
Line 271 Mentioning the most common tree species might help
Line 283 balks?
Figure (picture?) 1: please use the colour coding used here in subsequent figures; it seems to inverse.
Line 358 "Other farmers see the threat of losing subsidies for greening after planting trees" -- needs a bit more explanation
Line 377 "Climate change is not a sufficient driver, yet" strange subheading...

Reviewer 2 Report

The article is interesting insight into the perceptions of Czech farmers of agroforestry and climate change. A lot of research has been done here in order to present social dimension of agroforestry implementation in the country. The strength of the work is overview of history and policy background for farmers perceptions and decisions regarding climate change adaptability and trees in the agriculture landscape. This is very useful material for policy- and decision-makers. Considering the above, it is hard to understand why authors do not present hypothesis neither statistical characteristics of the collected data. The methods are not clearly presented, mixing focus groups, interviews and questionnaires in time line and scale. For example, regional seminars are identical to research focus group discussions? For better understanding of the section the graph presenting information flow and/or research objects connections would be useful. Also, geographical description of analysed regions (maps/photos of landscape) could be convincing. There is no characteristics of the sampled farming households (farm size, farmers age, sex, production type, ownership/tenancy, education level etc.). Thorough data analysis should be done (frequency, mean differences, significance etc.). Up to the authors, multifactor analysis or cluster analysis could be done as well in order to identify relatively homogenous groups of variables.

Some minor remarks:

please pay attention to:

naming stakeholders working with farmers (regional decision makers, not regions - see the abstract),

"the rest of the Europe" - what do you mean by that?

not generalizing European agriculture as industrial and biodiversity drop in terms of regional diversification,

"pests and diseases occurences" - those are natural processes

" there is still a great lack of knowledge about the functioning of agroforestry systems on the continental level" - did you mean "European level"?

The AFINET project has been already finished, so it is not new anymore.

Grassland description in terms of the habitat locations.

There are many problems or solutions, based on farmers statements that need to be verified by agronomists, since they seem to be wrongly named or interpreted (f.in. see the last two paragraphs of 3rd chapter)

The article needs careful checking style, grammar and to a lesser degree English language.